# Diet to Reduce the Metabolic Syndrome Associated with Menopause. The Logic for Olive Oil

**DOI:** 10.3390/nu12103184

**Published:** 2020-10-18

**Authors:** Juan José Hidalgo-Mora, Laura Cortés-Sierra, Miguel-Ángel García-Pérez, Juan J. Tarín, Antonio Cano

**Affiliations:** 1Service of Obstetrics and Gynecology, Hospital Clínico Universitario—INCLIVA, Av Blasco Ibáñez 17, 46010 Valencia, Spain; hidalgomorajj@gmail.com (J.J.H.-M.); cortes_lausie@gva.es (L.C.-S.); 2Department of Genetics, Faculty of Biological Sciences, University of Valencia, Burjassot, and INCLIVA, Av Blasco Ibáñez 17, 46010 Valencia, Spain; Miguel.Garcia@uv.es; 3Department of Cellular Biology, Functional Biology and Physical Anthropology, Faculty of Biological Sciences, University of Valencia, Burjassot, 46100 Valencia, Spain; Juan.J.Tarin@uv.es; 4Department of Pediatrics, Obstetrics and Gynecology, University of Valencia, Av Blasco Ibáñez 15, 46010 Valencia, Spain

**Keywords:** olive oil, metabolic syndrome, obesity, women, menopause, healthy ageing

## Abstract

The rates of metabolic syndrome are increasing in parallel with the increasing prevalence of obesity, primarily due to its concomitant insulin resistance. This is particularly concerning for women, as the years around menopause are accompanied by an increase in visceral obesity, a strong determinant of insulin resistance. A fall in estrogens and increase in the androgen/estrogen ratio is attributed a determining role in this process, which has been confirmed in other physiological models, such as polycystic ovary syndrome. A healthy lifestyle, with special emphasis on nutrition, has been recommended as a first-line strategy in consensuses and guidelines. A consistent body of evidence has accumulated suggesting that the Mediterranean diet, with olive oil as a vital component, has both health benefits and acceptable adherence. Herein, we provide an updated overview of current knowledge on the benefits of olive oil most relevant to menopause-associated metabolic syndrome, including an analysis of the components with the greatest health impact, their effect on basic mechanisms of disease, and the state of the art regarding their action on the main features of metabolic syndrome.

## 1. Introduction

The metabolic syndrome (MetS) consists of a cluster of risk factors that increase the risk of type 2 diabetes and cardiovascular disease (CVD) [1]. This cluster includes dysglycemia, increased blood pressure, lipid abnormalities as defined by hypertriglyceridemia and low high-density lipoprotein (HDL) cholesterol, and central obesity [2]. A conservative estimate is that around 100 million people may be affected worldwide, but the figure might be higher [3]. This makes the MetS a public health issue with a definitive impact on any healthy ageing strategy.

The prevalence of this syndrome is distributed heterogeneously worldwide, with geographic region, ethnicity, sex, age, socio-economic status, and education among the factors playing a role. It is believed that insulin resistance is the link underpinning the clustering of these risk factors [1]. Obesity, particularly central obesity, is also understood as a trigger because of its predisposing effect on insulin resistance [4].

## 2. Literature Search

We conducted a PubMed database search for publications between 1 January 2000 and 1 October 2020, pairing the term “olive oil” with “metabolic OR metabolic syndrome OR obesity OR central obesity OR weight OR waist OR blood pressure OR cholesterol OR triglycerides OR lipids OR insulin resistance OR diabetes OR menopause”. Only papers written in English or Spanish were considered, yielding a total of 7885 titles. The initial search considered the title, or title and abstract when the title raised uncertainty about the content of the paper, reducing the list to 251 articles. Systematic reviews and meta-analyses were included in the selection. We manually searched the reference lists of selected review papers to retrieve other citations of potential interest. Studies based on special populations (adolescents, transplant patients, pregnant women, etc.) were excluded. After cross-cleaning the lists, a total of 120 papers were chosen. (Figure 1).

## 3. Insulin Resistance in Menopausal Women

The rates of all forms of obesity are rising rapidly worldwide, and the problem is expected to worsen [5]. The association between excessive calorie intake and inactive lifestyle from an early age onwards has led to a global epidemic affecting both poor and rich countries. Women are affected by this obesity epidemic equally as [5], if not more so than, men [6].

Central obesity is defined by the abdominal accumulation of fat, which can be located mainly at the subcutaneous or visceral level, or both. Central visceral rather than subcutaneous obesity is a major determinant of insulin resistance, which has been considered a driver of detrimental outcomes in the MetS [7]. As with men, central obesity also confers increased risk in women, as shown in the 10-year follow-up of 156,624 postmenopausal women enrolled in the Women’s Health Initiative (WHI) cohort [8]. This finding is significant because unlike men, women have specific risk factors for central obesity. Indeed, women undergo dramatic hormonal changes at midlife, arising during the perimenopausal period, in which there is a significant decline in circulating estrogen levels [9]. Experimental and clinical studies concur that the fall in estrogens is associated with an increase in visceral fat [10,11,12]. Likewise, longitudinal population studies such as the Study of Women Across the Nation (SWAN) have confirmed that the odds of suffering from metabolic syndrome more than double in the years around the menopause [13]. This effect of menopause can undergo slight modifications according to ethnicity as a result of the different patterns of hormonal changes among women of different racial origins [14].

The potential contribution of hormonal changes other than those in estrogens has been studied in depth, with the case of polycystic ovary syndrome (PCOS) providing a good model [15]. Proof of the specific potential of hormones has been found in non-obese women presenting with PCOS, who also show an increased risk of MetS [16]. A relative increase in androgens vs. estrogens has been attributed a central role, although the issue is still a matter of some debate [17]. For example, androgen concentrations did not increase the risk for diabetes among overweight women who were already glucose intolerant in a secondary analysis of the Diabetes Prevention Program (DPP) and the Diabetes Prevention Program Outcomes Study (DPPOS) [18]. It might be that androgens distinct from testosterone, such as dehydroepiandrosterone-sulfate (DHEAS), could have a compensatory effect [19,20].

The androgen/estrogen imbalance also occurs at the time of menopause because of the fall in estrogens in the presence of a much lower decline in androgens (Figure 2). Confirmatory evidence has been obtained in a group of postmenopausal women whose testosterone levels were measured with an ultrasensitive method and their body composition and abdominal deposits with dual energy X-ray absorptiometry (DXA) and magnetic resonance imaging, respectively [21].

The potential influence of other factors cannot be excluded. For example, low levels of sex hormone binding globulin (SHBG) occurring when estrogen levels are low have been shown to be associated with the MetS and type 2 diabetes in both men and women [22,23]. More recent studies have gone further and have confirmed an increased risk for cardiovascular events. This was the conclusion from the observational cohort of 161,108 postmenopausal women enrolled in the Women’s Health Initiative (WHI) study, in which an inverse association between the serum levels of SHBG and the incidence of ischemic stroke was found [24].

It seems that the hormonal regulation of fat distribution is a strong variable affecting the differing fat distribution patterns between sexes, but much of the detail is still unknown. The possibility that a healthy diet may limit these menopause-dependent changes is an attractive hypothesis. The PCOS model is again illustrative, as recent clinical studies have shown that diet can worsen, when unhealthy [25], or improve, when healthy [26], hormonal and metabolic changes in the PCOS phenotype.

## 4. The Role of Healthy Nutrition

Lifestyle has been recommended as a first-line measure against MetS. Physical activity [27] and healthy nutrition are the two most widely promoted interventions [28]. A notable recent initiative is the EAT-Lancet Commission, which has underlined the need to foment healthy diet patterns that are respectful to local traditions and the environment. A healthy reference diet has been defined, with a high consumption of fruits and vegetables together with a reduction of processed meat or refined sugar as the main features [29]. The Mediterranean diet (MedDiet) has been thoroughly investigated and received worldwide recognition as one of the healthiest options [30]. Interestingly, the MedDiet has recently been recommended as a useful ally to manage women’s health needs during the menopause transition and after menopause [31].

Olive oil (OO) is among the most widely researched MedDiet components in both experimental models and clinical studies. Furthermore, indications in the literature suggest a role of OO in improving insulin sensitivity [32]. There is a dearth of studies specifically addressing the impact of OO on MetS disorders associated with menopause. However, there is considerable information on the action of OO on the mechanisms and the clinical features associated with the MetS. This information can be taken to better understand the effect of OO in limiting the development of MetS during menopause, as supported by a recent expert consensus [31]. In the coming sections, we will analyze the OO components with the greatest health impact, their effect on basic mechanisms of disease, and the state of the art regarding their action on the main components of MetS. Although there are four main OO subtypes (extra virgin, virgin, refined and pomace) [33], few studies discriminate by subtype, precluding us from considering them separately here.

## 5. Components in Olive Oil with a Health Impact

Olive oil includes a wealth of compounds, or compound families, in which unsaturated fat far exceeds saturated fats. Polyphenols form another group of compounds that contribute substantially to the health impact of OO.

### 5.1. Unsaturated Fat

Olive oil conforms to the recommendation put forth since the Seven Countries study [34] that saturated fat should be replaced by unsaturated fat of vegetable origin. The main OO components are oleic acid (70%), classed as a monounsaturated fatty acid (MUFA), and linoleic acid (15%), a polyunsaturated fatty acid (PUFA). Other unsaponifiable fatty acids may also be present in OO, depending on whether the variant is refined, virgin or extra virgin [35] (Figure 3).

### 5.2. Polyphenols

Polyphenols are a family of phytochemicals with a molecular structure containing phenol rings. Present in a wide variety of food sources, the members of this large family include flavonoids, phenolic acids, lignans and stilbenes, all of which exhibit both antioxidant and anti-inflammatory properties [36,37].

One predominant characteristic of polyphenols is their metabolism in the intestine, where a huge number (ranging between 100,000 and 200,000) of secondary compounds are generated with the intervention of local microbiota. The concentration of secondary metabolites falls sharply, from the mM to μM range in the original source to the nM range in the plasma [38].

The metabolic impact is rapid, as shown in a crossover study in which a reduction in foods containing polyphenols was already reflected in a change in biomarkers, such as the ratio of thromboxane A_2_ and prostaglandin I_2_, in the urine at the first control at 2 weeks [39].

The interest in polyphenols stems from findings of studies in other foods, for example, in several types of fruits, cocoa, etc., in which those compounds conferred a health protective effect [40,41]. Studies with OO have also supported this benefit.

#### Polyphenols in Olive Oil

The benefits of the polyphenols in OO have been shown in studies assessing either the effect of the whole family or the specific roles played by particular components.

Total polyphenol excretion in the urine was analyzed in an ancillary sub-study of the PREvención con DIeta MEDiterránea (PREDIMED) trial aimed at testing the efficacy of a MedDiet supplemented with extra-virgin olive oil (EVOO) or nuts, versus a control diet consisting of a recommended low-fat diet for primary CVD prevention [42]. Those of the 1139 participating individuals in the highest tercile of total urinary polyphenol excretion exhibited a lower plasma concentration of inflammatory biomarkers and significant improvement in cardiovascular indicators, namely, blood pressure and lipidograms [36], as reviewed in [43]. A meta-analysis has confirmed that polyphenol content is associated with an improved profile of several cardiovascular risk factors [44].

The specific role of certain polyphenols in OO, such as hydroxytyrosol (HxT) [45] and oleocanthal [46], have attracted particular attention (Figure 4).

The levels of 3-*O*-methyl-hydroxytyrosol, a urinary metabolite of HxT, have been reported as inversely correlated with the risk of CVD and overall mortality in elderly subjects [47]. As shown in this study, one important feature of HxT concerns its good bioavailability, in contrast with resveratrol, another molecule that has been ascribed healthy properties based only upon benefits in experimental terms, since in vivo bioavailability is poor [48].

Oleuropein is an ester of HxT that in experimental models has shown preventive effects in early-stage cancer [49]. Work on breast, thyroid and colorectal cell lines has shown anti-proliferative potential and pro-apoptotic effects [49].

## 6. Effect on General Mechanisms of Disease

Homeostasis in the human body is a direct reflection of functional status at the cellular level. Cellular damage may be the result of various mechanisms, including defects in elementary cellular functions such as respiration or nutrition, and the noxious actions of external agents.

Inflammation and oxidative stress are two intertwined basic mechanisms that play key roles in cellular damage during processes such as ageing [50,51] and disease [52]. Nutrition, which provides micronutrients, metal ions, and other cofactors, is considered to regulate oxidative stress and inflammation [53,54]. Additionally, microbiota have been recognized as a mechanism strongly sensitive to diet.

### 6.1. Inflammation and Oxidative Stress

Inflammation is an important factor in the onset and progression of sub-clinical phases, as well as in the occurrence of clinical events, in several diseases including CVD [52,55], osteoporosis [56], cancer [57], neurodegenerative disorders and Alzheimer’s disease, among others [58,59].

Oxidative stress, in turn, results from an imbalance between oxidant and antioxidant mechanisms. Reactive oxygen species (ROS) are small reactive molecules that regulate crucial biological processes. An excess of ROS generates reactions leading to DNA damage, the modification of proteins and the peroxidation of lipids. As has been noted for inflammation, these biological processes are involved in many diseases, such as atherosclerosis [60], type 2 diabetes [61] and others [62]. For example, a recognized effect of lipid peroxidation is an increase in oxidized LDL (oxLDL), a well-known pro-atherosclerotic factor [63]; this has led to proposed approaches to modulate oxidative stress (redox medicine) [53].

A close relationship exists between inflammation and oxidative stress [62]. For example, oxidative stress has been shown to intervene in the development and perpetuation of inflammation [64]. However, the opposite also occurs, as in the example of atherosclerosis, which is considered an oxidative response to inflammation [65].

#### The Impact of Olive Oil

The phenolic compounds in OO have shown a well-supported anti-inflammatory capacity [66,67,68], and clinical studies are confirmatory. The anti-inflammatory properties of the MedDiet have been demonstrated to be owing at least partly to OO [69]. Moreover, a study on high cardiovascular risk subjects showed that a higher intake of OO and nuts was associated with a reduction in several inflammatory markers, as exemplified by C-reactive protein, interleukin-6 and certain adhesion molecules [70]. A more recent randomized controlled trial (RCT) confirmed a differential impact of OO triterpenes [71].

There is also abundant information on the antioxidant effects of OO, which have been investigated using different experimental models [72]. The main MUFA in OO, oleic acid, is more resistant to oxidation than PUFAs. Phenolic compounds also show substantial antioxidant capacity [73,74,75], as is the case of HxT, which has demonstrated remarkable antioxidant potential in both in vitro and animal experiments—for a review, see [67,76,77]—as well as in healthy volunteers [78]. The free radical scavenging potential of HxT has been recognized by the European Food Safety Authority [79]. The EUROLIVE study has provided additional clinical support in finding that the phenolic content in OO was directly related to a reduction in heart disease risk factors [80].

Considerable antioxidant and anti-inflammatory potential has also been exhibited by oleocanthal, a phenolic compound responsible for the burning sensation at the back of the throat when consuming EVOO [81,82], and by other phenolic compounds [83,84].

### 6.2. Microbiota

The development of metagenomics technology has advanced the genomic study of microbes in the body. There is growing evidence linking obesity and type 2 diabetes with dysbiosis, a term describing alteration in the composition of intestinal microbiota. Dysbiosis is associated with changes in the intestinal barrier, which facilitate metabolite access to crucial organs such as the liver or fat. An overload of molecules such as lipopolysaccharides (LPS) and other endotoxins results in inflammatory processes, leading to clinically relevant conditions such as the aforementioned obesity [85].

Diet has an important effect on microbiota profiles and turnover [86,87]; for example, it has been shown that the MedDiet may change the gut microbiota, which has a knock-on clinical impact [88]. With regard to OO, an association of the intake of this oil with an increase in the biodiversity of the intestinal microbiota has been shown in rodent models [69]. As a general conclusion, the fatty acids in OO favor composition patterns with a higher prevalence of species that hinder dysbiosis. Polyphenols, by contrast, act as prebiotics and favor, among others, the genus *Bacteroidetes*, which are attributed a protective role against atherosclerosis [89]. Other microbiota modifications have been linked with changes in MetS features or other beneficial outcomes [90,91,92].

Clinical studies are still sparse and limited by the difficulty of establishing a causative role for the observed microbiota changes in the investigated outcomes. One small-scale RCT researching the effect of an OO-enriched biscuit found an increase in the output of the gut microbiota and metabolic changes suggestive of reduced oxLDL, although no real change in oxLDL could be detected [93]. Another small-sized RCT found that taking polyphenol-enriched OO for 3 weeks decreased oxLDL levels while increasing bifidobacteria and phenolic metabolite populations [94].

## 7. Impact on Metabolic Syndrome and Its Components

The protective role of OO against disease was addressed in a meeting of the International Olive Council [95], which highlighted mechanistic studies related to the action of polyphenols and fatty acids. The effect of OO on the MetS has been directly addressed in studies assessing the impact either on the MetS itself, or separately on each MetS component.

### 7.1. Metabolic Syndrome

There are a wealth of experimental studies, principally with cell cultures or rodent models, showing a role for polyphenols, mainly HxT, in improving MetS features [96,97], including some key ones such as adiposity and insulin resistance [77].

Intervention clinical studies have also yielded some evidence in this area. Supplementing a diet with EVOO, at least 4 tablespoons per day as in the PREDIMED study, was followed by a reversion of the MetS (control vs. EVOO hazard ratio (HR) = 1.35; 95% confidence intervals (CI): 1.15, 1.58) [98]. Some studies have used OO enriched with polyphenols, with mixed or inconclusive results [99] or with a reduction in certain features of the MetS, specifically, glycemia, blood pressure and LDL oxidation [100].

The current consensus is that more long-term RCTs are required to reach consistent conclusions [101]. Despite this, the international panel recommendation for MetS prevention and management through lifestyle included OO consumption at doses of 20–50 g/d along with the MedDiet [102].

### 7.2. Lipids

The EUROLIVE randomized trial found that the polyphenol content in OO was inversely associated with the total cholesterol/HDL ratio and triglyceride levels [80]. Other studies with virgin OO (30 mL/d) enriched with phenolic compounds and triterpenes have found increased HDL levels [103] and HDL functionality [104]. Further effects have been shown in smaller studies, including oxLDL reduction with polyphenol-enriched OO [94].

The differential impact of the polyphenols in OO has been analyzed in a meta-analysis including papers published up to December 2018. EVOO with a high phenolic content slightly reduced LDL cholesterol when compared with EVOO with low phenolic content (mean difference −0.14 mmol/L; 95% CI: −0.28, −0.01). Additional benefits were found for oxLDL, which showed an inverse dose–response relationship with the intake of phenolic compounds [105].

EVOO at the low dose of 10 g was also associated with a reduction in postprandial triglycerides in subjects with impaired fasting glucose levels [106]. Additionally related to this OO variant, PREDIMED has generated a list of sub-studies that have reported lipid changes associated with the EVOO-supplemented arm. Among these are a reduction in LDL atherogenicity, including resistance against oxidation, particle size, composition and cytotoxicity [107,108] and an increased cholesterol efflux capacity of HDL [108].

### 7.3. Blood Pressure

Some studies have associated the MUFA in OO with a reduction in vascular tone [109], so a decrease in blood pressure may be expected. Polyphenols are also involved in vascular dilation, according to data obtained from experiments with rodents [110]. Some peptides in OO have also been confirmed to exhibit anti-hypertensive activity through an angiotensin-converting enzyme inhibitory activity [111].

Clinical studies are still sparse, as was acknowledged in a systematic review that, based on data from only 69 subjects, concluded that systolic, but not diastolic, blood pressure was reduced by OO [112]. This review did not include the PREDIMED sub-study, which found a small reduction (mean: −2.3 mm Hg; 95% CI: −4.0, −0.5 for systolic, and mean: −1.2 mm Hg; 95% CI: −2.2, −0.2 for diastolic) in the EVOO-supplemented arm after 1 year in a subset of 235 subjects, with a mean age of 66.5 years, at high cardiovascular risk (85.4% with hypertension) [113]. It is unclear whether the effect may differ in individuals who are normotensive or free of other cardiovascular risk factors.

A more recent meta-analysis on the differential effects of distinct types of OO found an inverse, dose–response association between the phenolic compounds from OO and systolic blood pressure in a secondary analysis [105].

### 7.4. Body Weight and Waist Circumference

There is experimental evidence supporting a protective effect of the HxT in OO against adiposity [77,97], as reviewed in [96], an action that has also been shown for other polyphenols such as europein [114].

Zamora et al. conducted a systematic review and meta-analysis of RCTs with at least 12 weeks’ intervention in adults without CVD, analyzing papers published up to December 2016. Diets enriched in OO were more effective than control diets in weight reduction (−0.92 kg; 95% CI: −1.16–0.67), waist circumference reduction (−0.60 cm; 95% CI: −1.17, −0.04) and lowering body mass index (BMI) (−0.90 kg/m^2^; 95% CI: −0.91, −0.88) [115]. PREDIMED was included in the analysis, and the authors acknowledged that the large scale of that study influenced the conclusion.

A subsequent RCT in which OO was compared with coconut oil and butter could not detect any change in weight or waist circumference, but both the sample size (91 subjects) and the intervention duration (4 weeks) were limited [116].

### 7.5. Dysglycemia and Diabetes

As for the previous MetS components, the data in this area derive from work focused on the impact of polyphenols on different experimental models. There is a general consensus in that OO components, and particularly polyphenols, improve glycemic control [117], as reviewed in [118].

Clinical studies have been included in a meta-analysis of prospective cohort studies and trials. The findings show that the risk of diabetes in individuals in the highest OO intake category was lower than in the lowest one (relative risk (RR) = 0.84; 95% CI: 0.77, 0.92), and OO supplementation in subjects with type 2 diabetes was associated with a more pronounced reduction of HbA1c and fasting glycemia than in control groups [119].

## 8. Conclusions

The incidence of MetS is growing rapidly in women. The trend is probably influenced by the increase in the rates of obesity as a result of the presence of comorbid insulin resistance. These difficulties are exacerbated in women around the time of the menopause, when hormonal changes begin to promote central obesity. Given the association of metabolic syndrome with disease, this is a vital issue in any strategy focused on healthy ageing.

Guidance is therefore needed to overcome the problem, particularly during the menopause, given that this occurs at midlife, a crucial moment during which the sub-clinical phases of many non-communicable diseases often emerge. A healthy lifestyle, with nutrition as a vital component, needs to be implemented as a primary measure. For successful adoption and adherence, a healthy diet needs to be easy to follow and effective, two conditions successfully met by the MedDiet. The results of the above-presented data indicate that OO is a key food in the MedDiet that may prove especially helpful for women, particularly during this life stage. Experimental and clinical studies in the literature have been used as support. The clinical evidence, however, is limited by the observational nature of most studies.

## Figures and Tables

**Figure 1 nutrients-12-03184-f001:**
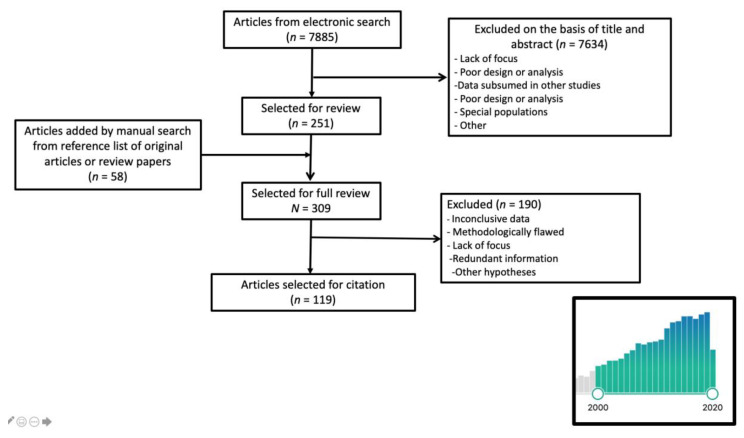
Literature search flowchart. The bars in the insert represent the trend in the numbers of papers published per year between 2000 and 2020. The year 2020 is incomplete because the search only included papers published until 1 October.

**Figure 2 nutrients-12-03184-f002:**
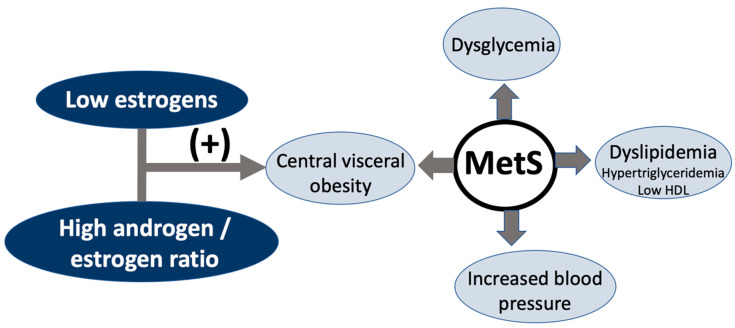
The metabolic syndrome (MetS) is defined as a cluster of four different risk factors, namely, dysglycemia, dyslipidemia, increased blood pressure, and central visceral obesity. While these affect both sexes, at midlife, women go through the menopause, which involves a rapid fall in estrogens and a very slow decline in androgens. Both the reduction in estrogens and the increase in the androgen/estrogen ratio have been attributed to promoting central obesity. Increased insulin resistance may then affect the other three factors in the cluster. HDL: high-density lipoprotein.

**Figure 3 nutrients-12-03184-f003:**
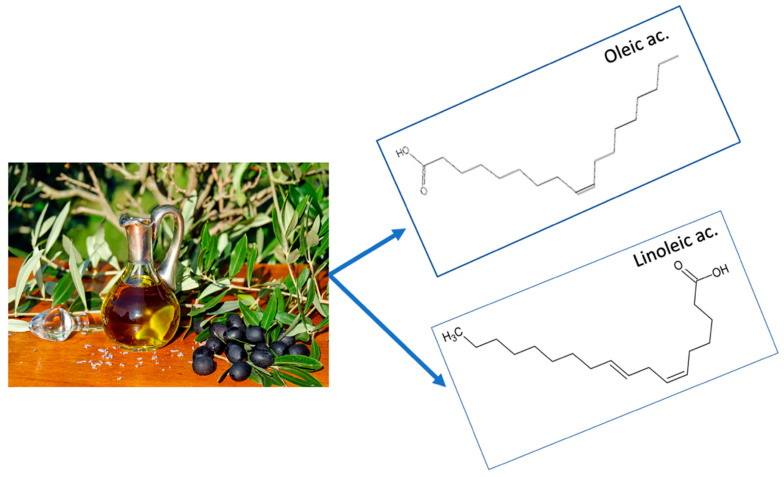
Olive oil is a source of unsaturated fat, whose two main components are oleic acid, a monounsaturated fat making up 70% of the total fat, and linoleic acid, a polyunsaturated fat representing 15% of the total fat content.

**Figure 4 nutrients-12-03184-f004:**
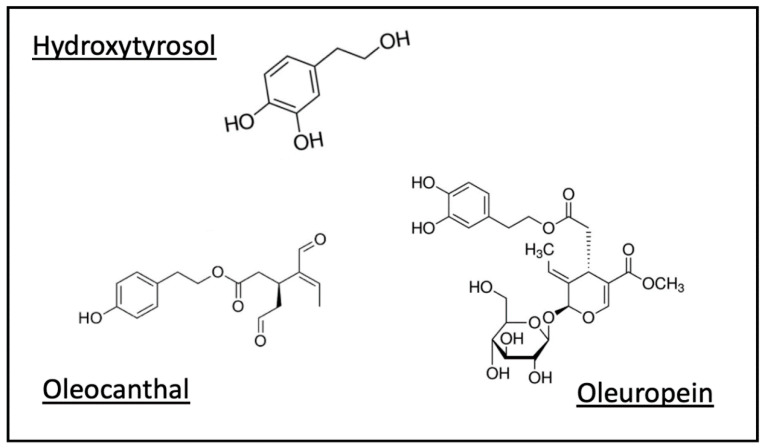
Polyphenols are a family of vegetal compounds characterized by phenol rings as part of their molecular structure. While the family includes a long list of compounds, current data can be found on the health benefits of some, with most available information centered on hydroxytyrosol, oleocanthal and oleuropein.

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
