# Peer review of "Diet to Reduce the Metabolic Syndrome Associated with Menopause. The Logic for Olive Oil"

_nutrients, 2020, doi:10.3390/nu12103184_

Round 1

Reviewer 1 Report

The authors are advised to cite more recent references and new findings on the proposed topic and make a comparison.

In Literature search section - It would be nice to add also a bar diagram to show the total number of publications during the mentioned period in order to highlight the research trend.

The limitations and disadvantages of the presented studies must be highlighted.

The English language and grammatical errors must be revised throughout the manuscript.

Author Response

We are grateful to the reviewer for his/her excellent work, which has translated into improvements that have substantially increased the merit of our manuscript. This is the point-by-point answer to the helpful comments.

Reviewer #1

  • The authors are advised to cite more recent references and new findings on the proposed topic and make a comparison.

Thank you for the remark.

Our literature search included papers up to August 22 2020 and yielded 7,826. We have now extended the search up to October 1st and yielded 7,885 papers. The analysis of the 59 additional papers yielded 2 further papers for review. We have also retrieved some more titles (11 additional papers) by hand-search from the reference lists of some of our already selected papers. Following suggestions of this and other reviewers, we have focused our attention on the hormonal basis for the metabolic changes across the perimenopausal period, and have finally selected 7 new papers for inclusion (see highlighted version). This has been the reason for increasing the total number of citations from 113 to 120. Specifically, 6 of the new papers have been published between 2017 and 2020.

The new information is intended i) to include the nuances imposed in the central obesity changes across menopause by ethnic differences, ii) to inform of the  sophisticated measurement of the accumulation of fat deposits by DXA and RM, iii) to strengthen the particulars of the androgen action,  and iv) to introduce data relative to the intervention of SHBG, a new and important factor, which is directly related with the estrogen changes. 

We agree with the reviewer in that this section needed strengthening and thank him/her again for the comment. The information from those updated references has been integrated in context and used to compare with the previous version, or to increase detail.

Also, we have also focused our attention onto the papers cited in section 7 (Impact on metabolic syndrome and its components), which includes references 88 till 120 (33 references in total). We have found that there is one reference from each 2008 and 2013, and three references  from each 2014 and 2015. The rest of the references (25 in total, a 76%) are papers published between 2016-2020. We hope the reviewer agrees in that the citations are up-to-date.

  • In Literature search section - It would be nice to add also a bar diagram to show the total number of publications during the mentioned period in order to highlight the research trend.

Thanks for the suggestion. We agree in that this information may help readers to better understand the research trend. We have done so by including an insert in Figure 1.

  • The limitations and disadvantages of the presented studies must be highlighted.

Thank you for the comment. We agree in that this is an important piece of information, which was poorly underlined in our previous version. We have now added a couple of sentences in the Conclusion (see lines 340-342) to stress that while we have presented evidence form basic and clinical studies, the final balance of harm or benefit is given by clinical studies and that, importantly, these are mainly observational.

  • The English language and grammatical errors must be revised throughout the manuscript.

Thanks again. Our manuscript was revised by an English editor before submission. This person has English as mother language. Moreover, she is an specialist in biomedical English edition. Indeed, she does this job for the manuscripts submitted from our Research Institute. She corrected and approved the final version, but of course, we are happy to implement any changes that might be suggested.

We hope we have addressed every point. Thanks again for the helpful comments.

Prof A. Cano on behalf of the other co-authors.

Reviewer 2 Report

Thank you for the opportunity to read the manuscript, which is well written and understandable.

  • I see a major problem in the title and abstract vs. the full text - it is obvious both in the title and abstract that you want to focus on the association of olive oil and metabolic syndrome with respect to menopause. However, in the full text there is very little information addressing menopause and the woman health. In my opinion, it nearly a general review of the association without an adequate focus on menopause for filling the title.
  • From line 236 on, you describe the effect of olive oil on metabolic syndrome and its components (again without a special focus on menopause or women...) - which, in my opinion, should be the main part of the article and at the same time form only a minority in scope. So I would recommend adding more information from the references, especially suitable into practice - for example amounts of olive oil consumed by subjects in the studies etc... Whereas some previous parts of the manuscript (e.g. lines 171-173) are too general and could be skipped.

Only details:

line 182 - Alzheimer’s DISEASE?

line 316 - "a healthy diet to be needs to be easy to follow and effective" - "to be" should be skipped?

Author Response

We are grateful to the reviewer for his/her excellent work, which has translated into improvements that have substantially increased the merit of our manuscript. This is the point-by-point answer to the helpful comments.

Reviewer #2.-

  • I see a major problem in the title and abstract vs. the full text - it is obvious both in the title and abstract that you want to focus on the association of olive oil and metabolic syndrome with respect to menopause. However, in the full text there is very little information addressing menopause and the woman health. In my opinion, it nearly a general review of the association without an adequate focus on menopause for filling the title.

Many thanks for the remark. We have now performed several actions:

  1. We have modified the tile. The actual one: Diet to reduce the metabolic syndrome associated with menopause. The logic for olive oil express, we hope, a more accurate announcement of what readers will find in the paper.
  2. We have substantially amplified the section dedicated to menopause, the hormonal background, and the updated information concerning the issue (see the new section 3). Following suggestions of this and other reviewers, we have focused our attention on the hormonal basis for the metabolic changes across the perimenopausal period, and have finally selected 7 new papers for inclusion (see highlighted version). This has been the reason for increasing the total number of citations from 113 to 120. Specifically, 6 of the new papers have been published between 2017 and 2020.

The new information is intended i) to include the nuances imposed in the central obesity changes across menopause by ethnic differences, ii) to inform of the  sophisticated measurement of the accumulation of fat deposits by DXA and RM, iii) to strengthen the particulars of the androgen action,  and iv) to introduce data relative to the intervention of SHBG, a new and important factor, which is directly related with the estrogen changes. 

We agree with the reviewer in that this section needed strengthening and thank him/her again for the comment. The information from those updated references has been integrated in context and used to compare with the previous version, or to increase detail.

  • From line 236 on, you describe the effect of olive oil on metabolic syndrome and its components (again without a special focus on menopause or women...) - which, in my opinion, should be the main part of the article and at the same time form only a minority in scope. So I would recommend adding more information from the references, especially suitable into practice - for example amounts of olive oil consumed by subjects in the studies etc... Whereas some previous parts of the manuscript (e.g. lines 171-173) are too general and could be skipped.

Thanks for the comment. We understand that a part of the answer to this comment has been already (the more extended attention to the metabolic changes along menopause and the menopausal transition) addressed in the previous point.

Concerning the pieces of information that may have a practical interest, like the olive oil dose, we have added so when available (see lines  266, 273, 276, and 285). We have been unable to do so for other references because they are reviews and include comments related with many papers.

Lines 171-173 can be skipped. Those lines correspond in the old version (that submitted and sent to reviewers) to a title and to two other lines in which there is a complete sentence and a truncated one. We are not sure we understand the suggestion. The whole section 6 is general and has been included to make readers understand better which general mechanisms may be affected by olive oil.

  • line 182 - Alzheimer’s DISEASE?

Thanks for the comment. Neurodegenerative diseases, including Alzheimer’s, are among the list of noncommunicable diseases that have an inflammatory background. This is supported by the 2 papers we have selected for citation, but there is extensive literature on the issue.

  • line 316 - "a healthy diet to be needs to be easy to follow and effective" - "to be" should be skipped?

Many thanks for the remark. Yes, “to be” should be, and has been skipped.

We hope we have addressed every point. Thanks again for the helpful comments.

Prof A. Cano on behalf of the other co-authors.

Reviewer 3 Report

This review article discussed the beneficial effect of olive oil on metabolic diseases that are associated with women menopause. Authors conducted a PubMed database search, systematically reviewed them, and well summarized the effects of olive oil on metabolic syndrome. Authors also discussed the ingredients of olive oil and its function. There are a few concerns that should be addressed.

 1. The title contains “menopause-associated metabolic syndrome”, but the Abstract and Conclusion are only focused on “obesity” in menopausal women. Authors should revise the abstract and conclusion sections.

 2. Authors should discuss the role of olive oil in estrogen level-related obesity and insulin resistance in detail, in the “3. Obesity and insulin resistance in menopausal women”.

 3. In the Fig. 2, the font and the size of letters should be revised. Full description is required for “High androgen/estrogen” in the figure.

Author Response

We are grateful to the reviewer for his/her excellent work, which has translated into improvements that have substantially increased the merit of our manuscript. This is the point-by-point answer to the helpful comments.

Reviewer #3.- 

This review article discussed the beneficial effect of olive oil on metabolic diseases that are associated with women menopause. Authors conducted a PubMed database search, systematically reviewed them, and well summarized the effects of olive oil on metabolic syndrome. Authors also discussed the ingredients of olive oil and its function. There are a few concerns that should be addressed.

  • The title contains “menopause-associated metabolic syndrome”, but the Abstract and Conclusion are only focused on “obesity” in menopausal women. Authors should revise the abstract and conclusion sections.

Thanks for the comment. We agree with the reviewer, and a part of the answer to this comment has been already addressed above (see first point for Reviewer #2. Moreover, we have drastically modified the initiation of the Abstract [omission of the 3 first lines of the old version and stress the interest on the metabolic syndrome and on insulin resistance, all that with a focus on menopause (see lines 18-21)]. Same in the Conclusion, where we have removed “obesity” as the first term and have modified the initial sentences to stress our focus on metabolic syndrome.

  • Authors should discuss the role of olive oil in estrogen level-related obesity and insulin resistance in detail, in the “3. Obesity and insulin resistance in menopausal women”.

Thanks for the comment. We agree in that this point cannot be missed in section 3 to give more internal coherence to the manuscript. We have now included a whole paragraph (see lines 156-162 ) and a new reference to accomplish that goal (De Bock et al., ref 32).

  • In the Fig. 2, the font and the size of letters should be revised. Full description is required for “High androgen/estrogen” in the figure.

Many thanks again. The size of the letters is heterogeneous on purpose, in order to underline the main areas in the figure, “MetS” and the hormonal determinants (“Low estrogens” and “High androg/estr ratio”).

Thanks a lot for pointing out the lack of definition of the acronym in the figure legend. We have now modified that window title and have written “High androgen/estrogen ratio”.

We hope we have addressed every point. Thanks again for the helpful comments.

Prof A. Cano on behalf of the other co-authors.

Round 2

Reviewer 2 Report

Dear authors,

Thank you for accepting my recommendations. We could probably have a discussion about the literature search; however, it is already a good work.

Please check the following lines (there might be minor mistakes): 

Line 93: “…the progression to diabetes of glucose intolerance, one component…”

Line 122 and further: “…the hormonal and the metabolic changes of the PCOS phenotype.”

Best regards!

Author Response

Thank you very much again for the new suggestions. We have rewritten the 2 selected sentences.

This is the new version:

"For example, androgen concentrations did not increase risk for diabetes among overweight women who were already glucose intolerant in a secondary analysis of the Diabetes Prevention Program (DPP) and the Diabetes Prevention Program Outcomes Study (DPPOS) [18]".

"The PCOS model is again illustrative, as recent clinical studies have shown that diet can worsen, when unhealthy [25], or improve, when healthy [26], hormonal and metabolic changes of the PCOS phenotype".

We hope the readability has increased now. We have also detected two minor errors in references 2 and 18, which have been corrected.

Many thanks again.

Best regards.
